# On the Health Benefits vs. Risks of Seaweeds and Their Constituents: The Curious Case of the Polymer Paradigm

**DOI:** 10.3390/md19030164

**Published:** 2021-03-19

**Authors:** João Cotas, Diana Pacheco, Glacio Souza Araujo, Ana Valado, Alan T. Critchley, Leonel Pereira

**Affiliations:** 1Department of Life Sciences, University of Coimbra, 3000-456 Coimbra, Portugal; jcotas@uc.pt (J.C.); diana.pacheco@uc.pt (D.P.); leonel.pereira@uc.pt (L.P.); 2Marine and Environmental Sciences Centre (MARE), Faculty of Sciences and Technology, University of Coimbra, 3001-456 Coimbra, Portugal; valado@estescoimbra.pt; 3Federal Institute of Education, Science and Technology of Ceará—IFCE, Campus Aracati, CE 040, km 137,1, Aracati 62800-000, Ceara, Brazil; glacio@ifce.edu.br; 4Department of Biomedical Laboratory Sciences, Polytechnic Institute of Coimbra, ESTeSC-Coimbra Health School, Rua 5 de Outubro, S. Martinho do Bispo, Apartamento 7006, 3046-854 Coimbra, Portugal; 5Verschuren Centre for Sustainability in Energy and the Environment, Sydney, NS B1P 6L2, Canada

**Keywords:** polysaccharides, health benefits, health risks, biomedical, polymer seasonal variation

## Abstract

To exploit the nutraceutical and biomedical potential of selected seaweed-derived polymers in an economically viable way, it is necessary to analyze and understand their quality and yield fluctuations throughout the seasons. In this study, the seasonal polysaccharide yield and respective quality were evaluated in three selected seaweeds, namely the agarophyte *Gracilaria gracilis*, the carrageenophyte *Calliblepharis jubata* (both red seaweeds) and the alginophyte *Sargassum muticum* (brown seaweed). It was found that the agar synthesis of *G*. *gracilis* did not significantly differ with the seasons (27.04% seaweed dry weight (DW)). In contrast, the carrageenan content in *C*. *jubata* varied seasonally, being synthesized in higher concentrations during the summer (18.73% DW). Meanwhile, the alginate synthesis of *S*. *muticum* exhibited a higher concentration (36.88% DW) during the winter. Therefore, there is a need to assess the threshold at which seaweed-derived polymers may have positive effects or negative impacts on human nutrition. Furthermore, this study highlights the three polymers, along with their known thresholds, at which they can have positive and/or negative health impacts. Such knowledge is key to recognizing the paradigm governing their successful deployment and related beneficial applications in humans.

## 1. Introduction

The growing demand for seaweed feedstock is noteworthy, particularly since 1990, reaching its peak in 2018, when 31.5 million tons fresh weight (FW) of seaweeds were sustainably cultivated (this is the latest year for which reliable data are available), while around 1 million tons FW of seaweeds were exploited from wild stocks [1]. Seaweeds’ rich nutritional profile (including phenolic compounds (e.g., phlorotannins), protein (e.g., phycobiliproteins), carbohydrates (e.g., alginates, fucoidans, ulvans, agars, and carrageenans), lipids (especially, ω-3 fatty acids), vitamins (in particular, A, B, C, D, E, and K and their pre-cursors) and essential minerals (e.g., calcium, iron, iodine, magnesium, and potassium)) has led to their incorporation in the daily diet of several Asian and European countries [2,3,4,5,6,7,8]. Furthermore, a significant amount of the total annual seaweeds feedstock is used by the global phycocolloid industry [1]. The main phycocolloids (i.e., algal-derived) or polysaccharides for these industries are agars and carrageenans (i.e., extracted from red seaweeds) and alginates (i.e., extracted from brown seaweeds) [7,9,10]. Polysaccharides (sugars) are highly valuable macronutrients, which are indeed abundant in seaweeds, as they act as structural components of the varied morphologies of the thalli. In fact, these molecules can represent up to half of thallus dry weight [11,12,13]. Seaweeds are a polyphyletic group, and across the >12,000 species, a wide range of polysaccharides are synthesized, differing within species (even at the level of alternation of generations (i.e., n vs. 2n) of the same species) and may vary due to biotic and abiotic stimuli [12,13,14,15,16]. Moreover, the selection of the methods of extraction and purification may also directly affect their yield and purity [17].

Such biomolecules are receiving a high degree of economic interest from several industries, including the feed, food, cosmetics and pharmaceuticals industries, due to their rheological (i.e., gelling/thickening) and increasingly wide range of biological activities. In particular, interest has blossomed in the food and pharmaceutical industries, because of many research studies on their bioactive properties, which include: antitumor [18], anticoagulant [19,20], anti-thrombotic [21,22], antiviral [23,24,25], immunomodulatory [26,27] and anti-fungal [28,29] activities, presented collectively and/or individually by these phycocolloids [30].

Currently, many seaweed polysaccharides are widely used in the food industry as stabilizers or emulsifiers for their gelling properties [31]. In fact, agar (E 406) [32,33], carrageenan (E 407), processed carrageenan (E 407a) [34,35] and alginate (E 401) [36,37] are authorized food additives and are generally recognized as safe (GRAS) for human consumption. Therefore, when food products with these seaweed polysaccharides (as additives) are ingested, they present properties similar to a dietary fiber, not least since humans do not have the required enzymes to break the glycosidic bonds of the long-chain carbohydrates [38]. Indubitably, humans need to metabolize sugars in order to fuel the body and the nervous system [39]. Hence, the inclusion of polysaccharides such as agar, carrageenan and alginate play important roles in human nutrition, since they promote satiety and intestinal function regulation, and enhance intestinal flora, consequently achieving higher nutrient absorption rates [11,16,40,41,42]. The consumption of algal polysaccharides provides several additional health benefits, such as regulation of glycemic index values and reduction of low-density lipoprotein (LDL) cholesterol [11,41,43,44].

Despite the many health benefits that seaweeds and their compounds provide, there are some concerns hampering some consumers from including them in their daily diet. Among these are iodine, metals, pesticides, antibiotics and or other noxious compounds (e.g., radionuclides), which some seaweeds may accumulate from coastal seawater [45,46,47]. Thus, the European Food Safety Authority (EFSA) and the competent North American authorities, e.g., the Food and Drug Administration (FDA), have established a threshold for the consumption of certain seaweeds and their components, through several health risk assessment studies [48,49,50]. It is key to note that only specific seaweeds are on these lists, while other species that are not in those lists are considered “novel food” and require a considerable amount of generational data and clinical trial evidence before being allowed in food supply chains.

The carrageenophyte *Calliblepharis jubata* is widely distributed through the European coastline, while the agarophyte *Gracilaria gracilis* and the alginophyte *Sargassum muticum* are widespread throughout the globe, and all are considered edible seaweeds [2]. All the forementioned species are perennial species in Portugal, meaning that they are present throughout the year [51]. Thus, they are species with potential for industrial exploitation.

Herein, this study aimed to analyze the seasonal variation of the polysaccharide content of three different seaweed species of agarophyte, carrageenophyte and alginophyte. It was further assessed whether the polysaccharide content met the threshold established by the competent authorities, thereby guaranteeing the safety for human consumption of the wild seaweeds. Additionally, the literature was reviewed in order to understand the positive and potentially negative effects of seaweed polysaccharides on human health.

## 2. Results

Throughout the seasons evaluated, the carrageenophyte *Callibepharis jubata* (Figure 1a) produced the lowest polysaccharide content, as compared to the agarophyte *Gracilaria gracilis* (Figure 1b) and the alginophyte *Sargassum muticum* (Figure 1c). Furthermore, different seasonal patterns were observed with respect to their polysaccharide profiles.

### 2.1. Red Seaweed Polymers

#### 2.1.1. Carrageenan Content and Identification

Regarding the red seaweed *C. jubata* (Figure 2), this species presented the lowest content during the autumn, with a concentration of 10.37 ± 0.416% DW. However, during the summer it was possible to extract 18.73 ± 2.382% DW carrageenan, which was the highest value during the observation period.

The FTIR-ATR spectrum (Figure 3) of the phycocolloids extracted from *C. jubata* showed bands at approximately 930 and 845 cm^−1^. However, an additional well-defined feature was visible at around 805 cm^−1^, indicating the presence of two sulfate ester groups on the anhydro-d-galactose residues, a characteristic band of the iota-carrageenan (Table 1) [52,53].

#### 2.1.2. Agar Content and Identification

The red seaweed *G. gracilis* produced the highest agar concentration (Figure 4) during the autumn, with 27.04 ± 2.684% DW. However, this did not differ greatly seasonally. Thus, its production was not statistically significantly different over the study period.

Agars differ from carrageenans, as they have an L-configuration for the 4-linked galactose residue; nevertheless, they have some structural similarities with carrageenans (Table 2). The characteristic broad band of sulfate esters, generally between 1210 and 1260 cm^−1^ (Figure 5), was much stronger in carrageenans than agars [54].

### 2.2. Brown Seaweed Polymers

#### Alginate Content and Identification

Regarding the non-indigenous seaweed *S. muticum* (Figure 6), the highest alginate concentration was observed during winter (e.g., 36.88 ± 2.953% DW).

The main polysaccharide found in the brown alga (*S. muticum*) was alginic acid, a linear copolymer of mannuronic (M) and guluronic acid (G). Different types of alginic acid present different proportions and/or alternating patterns of different guluronic (G) and mannuronic (M) units. The presence of these acids can be identified from their characteristic bands in the vibrational spectrum (Figure 7). The extracted colloid showed two characteristic bands: 806 cm^−1^, assigned to M units, and 788 cm^−1^, assigned to G units, suggesting the presence of similar amounts of mannuronate and guluronate residues (Table 3) [54,55].

## 3. Discussion

Seasonal variation in the concentration of different polysaccharides such as carrageenan, agar and alginate was investigated in order to better understand the impact of the season on the polysaccharide yield and quality, due to their economic value and applications. From an industrial management point of view, it is pivotal to assess the extraction yield and the costs associated with the production of these seaweed polysaccharides. The FTIR-ATR is a known method for characterizing and evaluating the overall quality and composition of the seaweed polymers, mainly on the basis of the concentration or modifications in the sulfate esters groups, which are among those groups that can vary seasonally [56,57,58,59]. The results of FTIR-ATR evaluation demonstrated similar spectra between seasons, thus revealing that the quality of the polysaccharide extracted differed only in terms of their yield in the *C. jubata* and *S. muticum*.

With respect to the carrageenophyte *C. jubata*, the optimum season to harvest this seaweed for the highest yield of carrageenan was mid-spring to the beginning of the summer. This red seaweed synthesizes lower carrageenan concentrations during the autumn and winter [59]. This observation was supported by other reports assessing seasonal yield variation in carrageenan on the Normandy (France) and Portuguese coasts [59,60]. It was found that in Normandy, carrageenan yields fluctuated from 15% (in winter) to 45% DW (at the end of the spring/beginning of the summer) [60]. On the Portuguese coast, the lowest carrageenan content was found during winter (i.e., 4% DW), but with a maximum yield during the spring (i.e., 40.4% DW, a 10 fold increase) [59]. Previous research has shown that *C. jubata* collected during spring 2020, at the same sampling site as the present study, produced a carrageenan yield of 28% DW [61]. The FTIR-ATR analysis of carrageenan from *C. jubata* was in concordance with the analysis of Pereira et al. [53], which detected iota-carrageenan with a low/residual content of kappa-carrageenan.

The increased accumulation of this biological reserve by *C. jubata* could be explained by the fact that the seaweed is typically from cold-temperate waters, so the increase of the surface seawater temperatures (SST) during the summer could be a stressor enhancing carrageenan biosynthesis [60].

The agarophyte *G. gracilis* is an opportunistic seaweed that is present in temperate-warm waters, and is already an important commercial source of agar, of which there is currently a global supply shortage [62,63,64]. Previous research has highlighted seasonal differences in the yield of agar from this species [65,66,67]. In agreement with the results of this study, *G. gracilis* collected on the Patagonian coast (Argentina) recorded the highest agar production during the summer and spring seasons (i.e., 41 and 30% DW, respectively) [65,66]. Using materials collected from the Venice Lagoon (Italy), an average agar yield of 25% DW was reported [67]. The lower agar yield during the spring/summer, can be attributed to the increased nutrient concentrations and lower turbidity and planktonic blooms that often characterize these seasons [67]. The FTIR-ATR analysis of the agar from *G. gracilis* demonstrated the presence of typical bonds for agar, with sulfate esters being evident. The observations are similar to the results obtained by Pereira et al. [54] with other agarophyte species (e.g., *Gelidium* spp.).

The alginophyte *S. muticum* is a brown seaweed introduced and well established in European and North American waters [62,68,69,70,71]. This seaweed can be used as a feedstock for alginate extraction, e.g., *S. muticum* harvested in Morocco produced a 25.6% DW yield at the beginning of spring [55,72]. The FTIR-ATR spectra of the alginate from *S. muticum* presented alginate peaks, but sulfate esters were also revealed, which could have been derived from sulfated polysaccharides such as fucoidan and laminarin [54].

The FTIR-ATR spectra were very similar between the seasons, demonstrating that the main factor was the quantity and not the quality. These results are in concordance with literature reports [54,59,64].

Seaweed polysaccharides, with a high molecular weight, are generally considered to be good dietary fibers. Specific applications of these are recognized as key players in human health and disease prevention [73]. These benefits are especially enhanced because there is an interplay with the gut microbiome at intestinal, as well as systemic, levels, resulting in homeostasis between the host and the microflora. Food intake can modify the microflora equilibrium positively or negatively, resulting in immunological, physiological, metabolic and even psychological effects. Consequently, the human diet can modulate health status: indubitably, we are what we eat [74,75,76].

In addition to their biological properties, seaweed polysaccharides also have innate properties that are very important for intestinal health; these include mainly the viscosity and the high potential for water-binding activity, which adjusts the transit time of food through the gut. Such properties are demonstrated to promote satiety and weight loss; additionally, they delay gastric emptying, thereby promoting glycemic control (i.e., reducing the incidence of diabetes). In the intestinal tract, all seaweed-derived polysaccharides are reported to enhance gut transit, maintaining regular stool bulking, and promoting beneficial alterations to the composition of the microbiome. Taken together, these benefits result in improved metabolization of volatile fatty acids (VFAs), which are also considered to be short chain fatty acids (SCFAs) by members of the microflora, promoting positive impacts in the gastrointestinal system, and thus resulting in the improved status of cardiometabolic, immune, bone, and mental health conditions [3,77,78,79,80].

It is now clear that various seaweeds have an interesting dietary fiber content, which can have a positive impact on the health status of production and companion animals, as well as on the health status of humans. Furthermore, this source is natural and uniquely different from crop and fruit plants. However, from this study and others, it is patently clear that not all seaweed polymers are “the same”. They have a structural function in the seaweed thalli and can be expected to vary seasonally. Hence, there is considerable need to quantify them, in order to ensure good intake without passing the intake limits. It has been shown that excessive consumption of dietary fiber can lead to negative impacts on human (and animal) health, e.g., recurrent symptoms of soft stools or diarrhea [3,77,78,79,81,82]. All good things should be taken in moderation. For instance, the study of Calvante et al. [83] demonstrated that a commercial powder of *Crassiphycus birdiae* at a low dosage, which is the recommend dosage intake of seaweed dried biomass supplement daily (5%) can induce reproductive toxicity and cellular damage when ingested with other chemicals. Thus, the full analysis of seaweed is required to fully understand the impact of seaweed compounds in the food industry and, more importantly, in the seaweed for direct intake [84,85], mainly with respect to metals (arsenic, cadmium, mercury, and lead) and other contaminants. In our study, the polymers were seasonally stable and the major differences in *C. jubata* and *S. muticum* were related to the yield.

The diversity of seaweed polysaccharides (and particularly their lower-molecular-weight oligomers) needs to be quantified, due to the negative effects that can arise if their cumulative dosage exceeds the limit of 25 g/day [82,86]. In this case, the consumption of wild harvested seaweeds would need to be limited according to their season of harvest (Table 1). Taking just three examples in the present study, *C. jubata* had the highest values in autumn; *G. gracilis* was the seaweed with the most consistent levels across the seasons; and *S. muticum* had the highest polymer levels during the winter. These observations are important in order to maximize the benefits of ingestion of particular types of seaweeds. This is because if the seaweed intake exceeds the recommended levels, the constituent polysaccharides, such as dietary fibers, can de-regulate the intestinal system, inducing bloating, abdominal pain, flatulence, loose stools or diarrhea, etc., as well as a reduction of blood glycemic values, which for diabetic patients, in particular, is a serious health risk [87].

However, due to the considerable diversity of seaweeds and their composition, the recommended daily intake for a generic “seaweed” is normally only 5 g DW/day, due to its high mineral/metal content; this was demonstrated by Milinovic et al. [88] as a result of the iodine content in seaweeds collected at Figueira da Foz, Portugal, which is a limiting factor in the seaweed intake. Due to the advice presented in the Recommended Dietary Allowance [89,90], it is necessary to standardize of the analyses applied to seaweed, especially for applications in the food industry [91]. Considering the examples in this study, the three species represent between 2 and 7% of the recommended daily intake (Table 4) [82,86].

Table 4 demonstrates that, overall, each of the seaweeds analyzed had a good dietary fiber content, which could be exploited commercially. In particular, *G. gracilis* did not exhibit a great deal of seasonal variability. However, *C. jubata* and *S. muticum* showed significant, although different, seasonal variations. From a commercial point of view, for the greatest benefit to the consumer, the seaweed raw materials need to be harvested in specific seasons in order that the level of polysaccharide content does not exceed the threshold of consumption. However, there is the need for a complete biochemical profile if the specific seaweed is to be consumed whole [90]. In particular, the macro- and trace elements present in the seaweeds need to be known due to their potential accumulation from the surrounding environment [88,92]. However, the effects due to the intake of whole seaweeds appear to be less when compared with the purified seaweed polysaccharide associated with water, milk, or prepared in a juice [93,94,95]. Consequently, ongoing research in this area is targeting applications of seaweed polysaccharides in novel foods with nutraceutical properties [6,96,97].

Anti-obesity effects have been described as being among the most beneficial attributes of seaweed polysaccharides for human consumption due to their fermentation in the intestinal tract, thereby reducing the microfloral/bile salt hydrolase activity, which is one theory behind this observed effect [96,98,99,100,101]. In this case, the microbiome composition was found to change to an augmented state, including populations of *Bifidobacterium*, *Bacteroides*, *Lactobacillus*, *Roseburia*, *Parasutterella*, *Fusicatenibacter*, *Coprococcus*, and *Fecalibacterium* colonies in in vitro experiments [96,98,99,100,101]. The nutritional values of the targeted seaweeds demonstrate a general fluctuation on the basis of the location at which the seaweed was collected, as demonstrated by Pacheco et al. [69]. Thus, the harvest site greatly influences the nutritional value, with carbohydrate yield being one of the principal variations (see Table 5). There is a lack of studies regarding the nutritional profile of *C. jubata*; however, there is a study by Araújo et al. [61] that characterizes its carbohydrate and lipidic profile.

However, as a food supplement, the safety of the dietary inclusion of seaweeds also needs to have various biochemical constituents checked before the alga can be made commercially available for regular human consumption [6,106]. Studies thus far have demonstrated that some wild harvested seaweeds, without thorough analysis of their nutrient and metal concentrations, can provoke negative impacts on health. However, there is little information available on this, relative to the more adverse pathologies (i.e., compared to those described above). This important topic is well described in the reviews of Cherry et al. [96] and Weiner [81], as well as Wierner and McKim [107], who demonstrated that within the daily recommend intake, there are indeed relatively low health risks to consumption. Nonetheless, major concerns have been expressed over seaweed polysaccharides present at low molecular weights, and poligeenan in particular. This has also been referred to as “degraded carrageenan”, and is not the natural chemical structure of the polysaccharide. Indeed, it can provoke harmful impacts, such as the powerful induction of inflammation. Intake of degraded carrageenan can happen when the legislation regarding polysaccharide preparation and usage in the food industry is not followed. Because of this, seaweed polysaccharide applications in the food industry are regulated, in order to guarantee the safety of the final product [6,32,34,36,81]. Aside from the general considerations regarding the safety of seaweed polysaccharides, there is ongoing debate arising from several in vitro and in vivo assay reports [108,109,110,111,112,113,114,115]. Unfortunately, there is still a lack of standard methods, and there are only a few in vivo assays with fully characterized seaweed polysaccharides [3,96]. This was demonstrated by Kumar and Sharma [116], where several deaths and illnesses that had been attributed to consumption of seaweeds were found to be mainly due to wild harvesting at unsuitable (polluted) sites, unreasonably high consumption, and the noted presence of highly potent secondary metabolites/toxins (some microalgal bloom related).

However, despite these negative reports, which occur only rarely, judicial (i.e., in moderation) consumption of seaweed polysaccharides have overall positive effects on several aspects of human health. These polymers work as nutraceutical compounds, thereby promoting human welfare and health. Indeed cautious and responsible consumption of seaweeds is no different from that of other terrestrial and marine food sources [117]. Taken alone, isolated seaweed polysaccharides have demonstrated numerous interesting properties for use in pharmaceutical and medical applications. In this regard, several specific seaweeds and their components are already in use commercially, while others are still in the research and development stage [117]. As commercial examples, the use of alginate in wound dressings, carrageenans in antiviral solutions, and agar in encapsulation of pharmacological drugs are all impressive [118,119,120,121,122]. In experimental development, selected seaweed polymers are being targeted mainly for the development of new hydrogel-based models for various human conditions, such as tumor or cardiovascular diseases, in order to provide more comprehensive models with which to understand drug and human cell interactions without using in vivo animal models, thereby providing more accurate/predictable responses [123]. This approach also enhances the development of new hydrogels for tissue engineering, where seaweed polymers have demonstrated good results in the early stage of development [124,125,126].

Seaweed polysaccharides can be applied as cosmetic ingredients, being used as gelling agents, thickeners, protective colloid emulsifier and stabilizer agents in hand lotions and liquid soap, deodorants, makeup, exfoliant, cleanser, shaving cream, facial moisturizer/lotion or in creams for acne and anti-aging care [127]. Similarly polysaccharide formulations can also be used in skin protection cosmetics to combat dermatitis, psoriasis, eczema, and dryness [128].

Carrageenans are one of the most bioactive polysaccharides from seaweeds; their chemical structure allows the formation of hydrogels, thereby allowing them to be used in anti-viral and anti-bacterial ingredients in various formulae [128,129]. There are compelling reasons for the use of these compounds, given the high levels of safety, efficacy and biocompatibility reported, in addition to their being biodegradable and non-toxic [118,130]. Furthermore, ancient records show that carrageenan has been used as a traditional medicine to ameliorate coughs and the common cold. These “old wives’ tales” have been supported more recently by *in vitro* and *in vivo* assays using animal models. This functionality is mainly derived from the actions of carrageenans in inhibiting the aggregation of blood platelets (i.e., anticoagulant activity) [74,131]. Various carrageenans have other demonstrable bioactivities such as anti-tumor, anti-viral and immunomodulation activities [116,132], which are already being exploited commercially. The anti-viral mechanism is based on blocking the entrance of viral particles into the cell. Good results have been demonstrated against the herpes simplex virus type 1 and type 2, HIV-1, and the human rhinovirus [133,134]. These anti-viral activities have mainly been observed in iota-carrageenan (which is the carrageenan type produced by *C. jubata*) [53,59,133]. However, in pharmaco-dynamics, carrageenans that are harmful for human consumption (specifically, in the form of oligo-carrageenans or poligeenans) are regularly used as a pro-inflammatory factor in diverse *in vitro* and *in vivo* assays, due to the high inherent bioactivity when degraded to a low molecular weight [117]. Oligo-carrageenans can also be used to induce pleurisy, paw edema and ulceration in animal models, and as such, they are used as tools for medical research [135].

In contrast to carrageenans, agars and alginates are not recognized as bioactive molecules—instead they are seen as excellent polymers with reduced bioactivities (i.e., they are biologically inert) that can be inserted and used as a barrier/encapsulation to stabilize active ingredients and develop new biomedical and pharmaceutical methods and techniques [129,136]. Agar is used in pharmaceutical products such as a bulking and suspension ingredients for medical solutions, anti-coagulant agents, and laxatives in capsules and tablets [132,137]. Alginates are perhaps the most used seaweed polysaccharides in medical and pharmaceutical products already on the market, namely in wound and battle dressings, and also in wound-healing products in the form of hydrogels [34]. Alginates, when used in the biomedical and pharmaceutical areas, are linked to cations, such calcium, sodium or magnesium, to produce a biopolymer with no bioactivity and low toxicity that is easy to manipulate so as to permit the development of hydrogels for tissue regeneration, as well as application in other areas such as in burn or diabetic wound-healing dressings [118].

However, seaweed polysaccharides have been further explored in drug delivery systems, where the polymers have demonstrated features such as natural biocompatibility, variation of viscosity and gelation conditions, low toxicity, low-cost polymers, and biodegradability, with easy adaptation and manipulation for the assembly of polymer-derivatives with new physical characteristics [117,118,138]. Seaweed-derived polysaccharides have adaptable swelling properties that respond to temperature modifications, which is important for on-demand and time-dependent modulation of drug release [139]. In the post-rheology, pharmaceutical and medical arenas, seaweed polysaccharides must have a high level of purity in order to reduce the impact of potential inclusion of impurities in the application of polymers in products and solutions, permitting clean application without any associated health risks or hazards [140].

## 4. Materials and Methods

### 4.1. Reagents

The reagents used for carrageenan extraction, i.e., methanol, acetone, ethanol and sodium hydroxide, were acquired from the suppliers José Manuel Gomes dos Santos, Lda., Odivelas, Portugal; Ceamed, Lda., Funchal, Portugal; Valente e Ribeiro. Lda., Belas, Portugal and Sigma-Aldrich GmbH, Steinheim, Germany, respectively.

The reagents used for alginate extraction, i.e., sodium carbonate and hydrochloric acid, were purchased from Fisher Chemical, Leicestershire, United Kingdom.

### 4.2. Seaweed Collection

Seaweeds belonging to the Rhodophyta, i.e., *Gracilaria gracilis* (Stackhouse) Steentoft, L.M. Irvine & Farnham 1995, and *Calliblepharis jubata* (Goodenough & Woodward) Kützing 1843 and the Ochrophyta, i.e., *Sargassum muticum* (Yendo) Fensholt, were collected during low-low tide the intertidal of Buarcos Bay, located in Figueira da Foz, Portugal (40°10′18.6″ N, 8°53′44.4″ W), Portugal. The seasonal sampling was conducted during 2020 (see Table 6) from sites with well-established seaweed populations without epiphytes or degradation visible to the eye. The specimens were collected from three different tidal pools at the same height on the shore. Approximately 100 g FW of *C. jubata,* 300 g FW of *G. gracilis*, 200 g FW of *S. muticum* were collected. After harvesting, each species was kept separately in plastic bags, inside a cool box, and transported to the laboratory (50 min from the harvest location). Firstly, the thalli were washed with filtered seawater to remove sand and other detritus. Thereafter, the samples were washed with distilled water, aiming for the removal of excess salts caused by the previous washing process and then placed on plastic trays and placed into an air-forced oven (Raypa DAF-135, R. Espinar S.L., Barcelona, Spain) at 60 °C, for 48 h. The dried algae were finely ground to make uniform (≤1 mm) samples with a commercial mill (Taurus aromatic, Oliana, Spain) and then stored in a box with silica gel to reduce humidity, in the dark, at room temperature (±24 °C).

### 4.3. Polysaccharide Extraction

#### 4.3.1. Agar

Agar extraction was based on the technique reported by Li et al. [141], with adaptations. The extraction was performed in triplicate, using 20 g of dried seaweed and 600 mL of distilled water. Afterwards, the solution was placed in an electric pressure cooker (Aigostar 300008IAU, Aigostar, Madrid, Spain) at a temperature of 115 °C with an air pressure of 80 Kpa, for 2 h. The solution was hot filtered, under vacuum, in a Buchner funnel, through a cloth filter. The extract was then vacuum filtered with a Goosh funnel (porosity G2). At room temperature, the filtrate was allowed to gel, frozen overnight and thawed. The thawed gel was finally dried (60 °C, 48 h) in an air-forced oven (Raypa DAF-135, R. Espinar S.L., Barcelona, Spain).

#### 4.3.2. Carrageenan

The extraction of carrageenan was carried out in triplicate, according to the method defined by Pereira and van de Velde [142]. To remove the organic-soluble fraction, 1 g of milled seaweed was pre-treated with an acetone:methanol (1:1) solution at a final concentration of 1% (*m*/*v*) for 16 h, at 4 °C. The liquid solution was decanted, and the seaweed residues were collected and dried at 60 °C in an air-forced oven (Raypa DAF-135, R. Espinar S.L., Barcelona, Spain).

Dried samples were immersed in 150 mL of NaOH (1 M) in a hot water bath (GFL 1003, GFL, Burgwedel, Germany), at 85–90 °C, for 3 h. The solutions were hot filtered, under vacuum, in a Buchner funnel with a cloth filter. The extract was vacuum filtered with a Goosh funnel (porosity G2). Under vacuum, the extract was evaporated (rotary evaporator model: 2600000, Witeg, Germany) to one-third of the initial volume. The carrageenan was precipitated by adding twice the final volume of 96% ethanol. The polysaccharide was washed with 96% ethanol for 48 h at 4 °C and dried in an air forced oven (60 °C, 48 h).

#### 4.3.3. Alginate

The extraction of alginic acid was performed in triplicate, employing the adjusted method of Sivagnanavelmurugan et al. [143]. Milled seaweed was added to a solution of HCl at 1.23% (1:30 v:v) and kept at room temperature for 48 h. The solution was filtered under vacuum with a Goosh funnel (porosity G2). The residue was rinsed with distilled water two to three times. The residue was submitted to an alkali extraction in 2% sodium carbonate for 48 h. The solution was filtered under vacuum through a cloth filter supported in a Goosh funnel (porosity G2), to remove any residues from the alginate solution. This process was followed by the addition of a solution of 37% HCl to the filtrate, producing the alginic acid precipitation (1 mL of 37% of HCl: 30 mL of the final solution). The precipitate was separated by centrifugation (4000 rpm, for 15 min) and then dried in an air forced oven (60 °C, 48 h).

### 4.4. Carbohydrate Characterization

#### Polysaccharide Analysis

For Fourier-Transform Infrared Spectroscopy–Attenuated Total Reflection (FTIR-ATR) analyses, the dried polysaccharides were powdered using a commercial mill, and then subjected to direct analysis. FTIR-ATR spectra were recorded on a Perkin Elmer Spectrum 400 spectrometer (Waltham, MA, USA), with no need for sample preparation, since these assays only require dried samples [52,53,54]. All spectra presented are the average of two independent measurements from 650–1500 cm^−1^ with 128 scans, each at a resolution of 2 cm^−1^. The FTIR-ATR spectra in the manuscript were performed with the polymers extracted in the autumn.

### 4.5. Statistical Analyses

The statistical analyses were performed using Sigma Plot v.14. This included an ANOVA analysis to assess statistically differences between the extraction yields. Holms-Sidak multiple comparison analysis was used after the rejection of the ANOVA null hypothesis, to discriminate any differences. The analyses were considered statistically different when *p*-value < 0.05. Error bars are the standard deviation of the mean.

## 5. Conclusions

The seaweeds analyzed in this study demonstrated that wild harvested materials can indeed vary in terms of polysaccharide yield. Such variance could significantly change the nutritional value/properties on a seasonal basis. Therefore, the direct intake of seaweeds should be carefully analyzed.

Seaweed-derived polymers (polysaccharides) as food sources/ingredients are compared to dietary fiber due to their high molecular weight and because algal polysaccharides are not digestible compounds, being an important nutraceutical for good gastrointestinal functioning. However, if the polysaccharide is degraded (by over hydrolysis), the low-molecular-weight fractions can have negative impacts on human health. Likewise, over-dosage/consumption may be an issue, and thus moderation in all things is a key.

However, for the commodity food sector, there is a need to guarantee similar nutritional values in all supplies, independent of the season. In this context, seaweed cultivation can present a solution for controlling seaweed food safety; this is something in which Asian countries are already well practiced [144,145,146,147,148], and which Western countries need to learn and adapt to.

In the future, long-term assays should be conducted in different years to understand any fluctuations that may occur. Thus, seaweed cultivation (on land or in the sea) may provide more homogenous raw materials. In the nutraceutical/biomedical field, there is a need to understand the digestive part of the polymers in order to provide greater security for the consumption of seaweed polymers. As demonstrated, there is a need to understand the ecological factors affecting seaweed biomass in order to obtain safe and high-quality polymers to support their many applications in the food industry.

## Figures and Tables

**Figure 1 marinedrugs-19-00164-f001:**
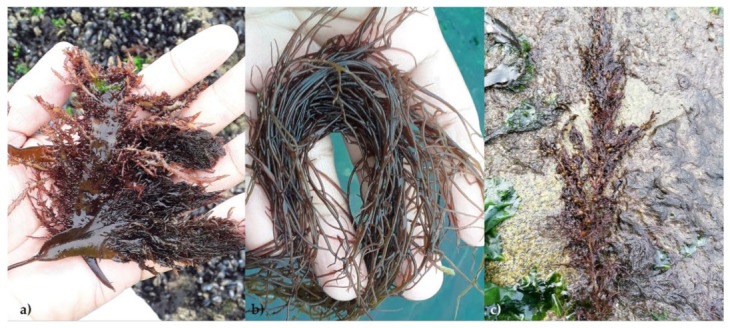
Three seaweeds studied at the collection site (Figueira da Foz, Portugal): (**a**) *Calliblepharis jubata* (Rhodophyta—carrageenan-bearing); (**b**) *Gracilaria gracilis* (Rhodophyta—agar bearing); (**c**) *Sargassum muticum* (Phaeophyta—alginate-bearing).

**Figure 2 marinedrugs-19-00164-f002:**
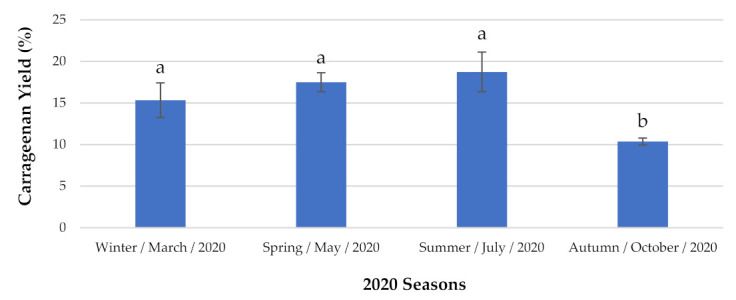
Carrageenan content analyzed seasonally. The extraction yields are expressed as mean ± standard deviation (*n* = 3). ^a,b^ The same letters indicate no significant differences at the *p*-value < 0.05 level.

**Figure 3 marinedrugs-19-00164-f003:**
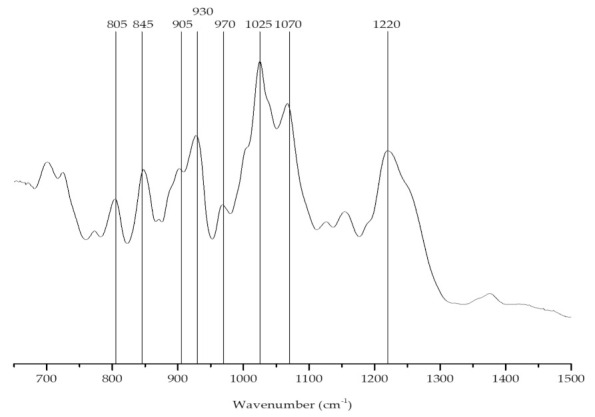
FTIR-ATR spectrum of the carrageenan extracted from *Calliblepharis jubata*.

**Figure 4 marinedrugs-19-00164-f004:**
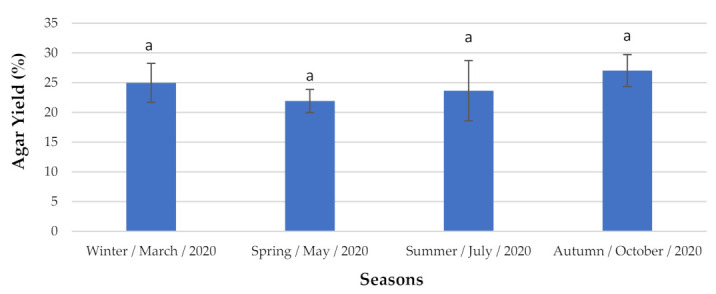
Agar content analyzed seasonally. The extraction yield results are expressed as mean ± standard deviation (*n* = 3). ^a^ The letters indicate no significant differences at the *p*-value < 0.05 level.

**Figure 5 marinedrugs-19-00164-f005:**
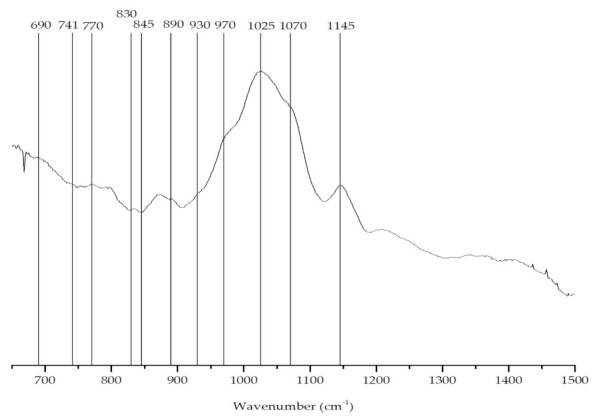
FTIR-ATR spectrum of the agar extracted from *Gracilaria gracilis.*

**Figure 6 marinedrugs-19-00164-f006:**
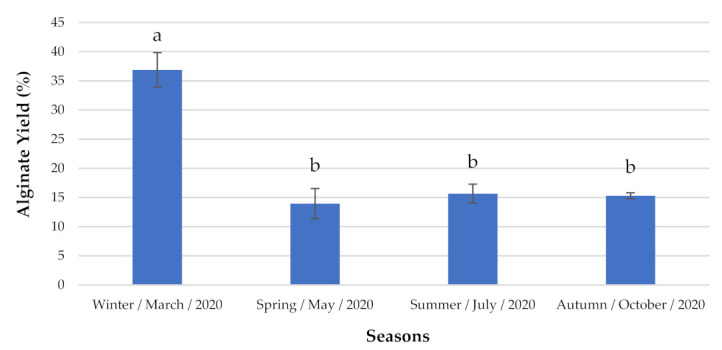
Alginate content analyzed seasonally. The extraction yields are expressed as mean ± standard deviation (*n* = 3). ^a,b^ Similar letters indicate no significant differences at the *p*-value < 0.05 level.

**Figure 7 marinedrugs-19-00164-f007:**
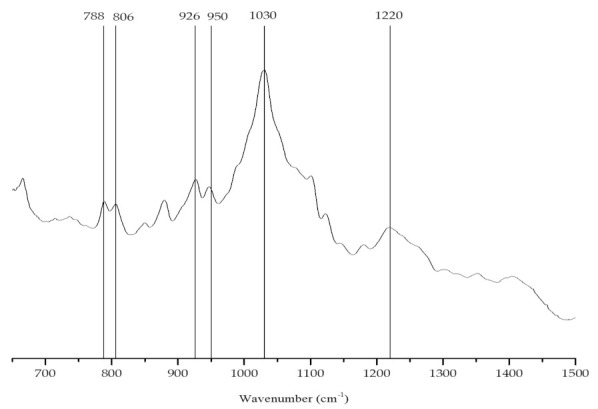
FTIR-ATR spectrum of the alginate extracted from *Sargassum muticum*.

**Table 1 marinedrugs-19-00164-t001:** FTIR-ATR band identification and characterization of the red seaweed *Calliblepharis jubata* polysaccharides (carrageenan), based on the literature [53,54].

Wave Number (cm^−1^)	Chemical Group	Letter Code
805	C–O–SO_3_ on C2 of 3,6-anhydrogalactose	DA2S
845	d-galactose-4-sulfate	G4S
905	C–O–SO_3_ on C2 of 3,6-anhydrogalactose	DA2S
930	C–O of 3,6-anhydrogalactose (agar/carrageenan)	(DA)
970–975	Galactose	G/D
1025	Sulfate esters	S=O
1070	C–O of 3,6-anhydrogalactose	DA
1240–1260	Sulfate esters	S=O

**Table 2 marinedrugs-19-00164-t002:** FTIR-ATR band identification and characterization of the red seaweed *Gracilaria gracilis* polysaccharides (agar), based on [53,54].

Wave Number (cm^−1^)	Chemical Group	Letter Code
690	3,6-anhydro-l-galactose (agar)	Agar
741	C-S/C-O-C bending mode in glycosidic linkages of agars	Agar
790	Characteristic of agar-type in second derivative spectra	Agar
805	C–O–SO_3_ on C2 of 3,6-anhydrogalactose	DA2S
845	D-galactose-4-sulfate	G4S
890–900	Unsulfated b-d-galactose	G/D
930	C–O of 3,6-anhydrogalactose (agar/carrageenan)	(DA)
1012	Sulfate esters	S=O
1070	C–O of 3,6-anhydrogalactose	DA
1100	Sulfate esters	S=O
1240–1260	Sulfate esters	S=O

**Table 3 marinedrugs-19-00164-t003:** FTIR-ATR band identification and characterization of the brown seaweed *Sargassum muticum* polysaccharides (alginate), based on [54].

Wave Number (cm^−1^)	Chemical Group
788	Mannuronic acids residues
806	Guluronic acids residues
1020	Alginic acid
1232	Fucoidan
930–950	C-O stretching vibration of uronic acids

**Table 4 marinedrugs-19-00164-t004:** Thresholds of daily consumption of seaweeds based on their polysaccharide content.

Season	*C. jubata* (g of Dried Seaweed for 25 g of Dietary Fiber)	*C. jubata* (g of Dietary Fiber for 5 g of Dried Seaweed)	*G. gracilis* (g of Dried Seaweed for 25 g of Dietary Fiber)	*G. gracilis* (g of Dietary Fiber for 5 g of Dried Seaweed)	*S. muticum* (g of Dried Seaweed for 25 g of Dietary Fiber)	*S. muticum* (g of Dietary Fiber for 5 g of Dried Seaweed)
Winter	163.04	0.77	100.11	1.25	67.79	1.84
Spring	142.93	0.86	114.07	1.10	179.14	0.70
Summer	133.48	0.94	105.70	1.18	159.57	0.78
Autumn	241.15	0.52	92.45	1.35	163.29	0.77

**Table 5 marinedrugs-19-00164-t005:** Range of nutritional values of selected seaweeds analyzed around the world (% DW).

Seaweed Species	Protein	Lipid	Carbohydrate	Ash	Ref.
*G. gracilis*	5.83–20.2	low	9.52–68.13	6.78–24.78	[102,103,104,105]
*S. muticum*	4.64–22	0.12–3.2	27.9–69	13.2–26.4	[69]

**Table 6 marinedrugs-19-00164-t006:** Seaweed collection data.

Season	Date	Water Temperature (°C)	pH	Salinity (ppm)	Conductivity (µS/cm)	ORP (mV)	O_2_ (%)
Winter	9 March 2020	14.13 ± 0.08	7.8 ± 0.09	35.12 ± 0.17	42146 ± 130.07	111.07 ± 5.27	113.2 ± 6.12
Spring	27 May 2020	17.15 ± 0.07	8.55 ± 0.06	35.60 ± 0.10	44149 ± 53.68	53.68 ± 27.33	94.775 ± 6.41
Summer	20 July 2020	17.94 ± 0.62	8.24 ± 0.19	36.20 ± 0.05	54595 ±184.33	184.33 ± 16.87	35.16 ± 3.68
Autumn	19 October 2020	14.49 ± 0.08	8.06 ± 0.10	35.78 ± 0.12	54065 ± 83.56	83.56 ± 7.62	88.77 ± 4.00

## Data Availability

Data available from authors.

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
