# Peer review of "On the Health Benefits vs. Risks of Seaweeds and Their Constituents: The Curious Case of the Polymer Paradigm"

_marinedrugs, 2021, doi:10.3390/md19030164_

Round 1

Reviewer 1 Report

The work entitle "On the health benefits vs. risks of seaweeds and their constituents: The curious case of the polymer paradigm" developed by Cotas and coauthors is on the focus of the Marine Drugs. Some suggestions have been proposed:

In the section: "2.1.2. Agar content and identification" authors wrote:
"The red seaweed G. gracilis synthesized the highest agar concentration (Figure 3) dur-ing the autumn, with 27.04±2.684% DW. However, the seasonality was not statistically significant different over the study period."

The authors should revise this sentence, in my opinion the numer of the figure is not correct, should be Figure 4.

On the other hand, in the discussion section, in a paragrah we can read: "Consequently, the human diet can modulate health status, indubitably, we are what we eat [76–78]!"

I suposse the "!" at the final of the sentence is a mistake.

In the section "4.2.1. Agar" the authors write: "The extract was then vacuum filtered with a Goosh 2 silica funnel"

I suggest revise this sentence, I think is a Gooch funnel.

Author Response

Reviewer 1:

Comment 1: The work entitled: "On the health benefits vs. risks of seaweeds and their constituents: The curious case of the polymer paradigm" developed by Cotas and coauthors is within the focus of the Marine Drugs. Some suggestions have been proposed:

In the section: "2.1.2. Agar content and identification" authors wrote:

"The red seaweed G. gracilis synthesized the highest agar concentration (Figure 3) during the autumn, with 27.04±2.684% DW. However, the seasonality was not statistically significant different over the study period."

The authors should revise this sentence, in my opinion the number of the figure is not correct, should be Figure 4.

On the other hand, in the discussion section, in a paragraph we can read: "Consequently, the human diet can modulate health status, indubitably, we are what we eat [76–78]!"

I suppose the "!" at the final of the sentence is a mistake.

In the section "4.2.1. Agar" the authors write: "The extract was then vacuum filtered with a Goosh 2 silica funnel"

I suggest revise this sentence; I think is a Gooch funnel.

Answer 1: The authors acknowledge the kind advice of the reviewer. All the suggestions were addressed and corrected.

Reviewer 2 Report

The manuscript entitled “On the health benefits vs. risks of seaweeds and their constituents: The curious case of the polymer paradigm” by Cotas et al. comprises the necessary elements of scientific novelty. 

Minor comments

The first two points in the abstract should be simple for readers.

In the Abstract… Authors focused mostly on the background of macroalgae, and need to write details about their findings also. Please rewrite the abstract.

The introduction part needs to be improved towards connected the para to para.

Can you please describe on what basis the authors selected the three examples? What is the current status of the three examples, in research?

Did season really affected the content... however, you showed no significant differences in Figure 2. Did you check the quality? Please write.

Please write clearly about your findings shown Figures 3, 5 & 7: In which season content analysis was done.

In the discussion section, where you discussed the gut microbiome, please also write about the nutritional values of seaweeds in brief.

The content benefits humans through various health benefits… as per the given title, what about the risk factors...??

In this study, you restricted yourself up to FTIR analysis. Improve the discussion with other supportive research data.

The outcome of the research in the abstract is not the same as that at the end of the Conclusion section. Please update. Conclude your findings and their future directions….

I would recommend the publication of this manuscript after addressing minor changes.

Author Response

Reviewer 2:

Comment 1: The manuscript entitled: “On the health benefits vs. risks of seaweeds and their constituents: The curious case of the polymer paradigm” by Cotas et al. includes the necessary elements of scientific novelty.

Answer 1: The authors acknowledge the kind words of the reviewer.

Minor comments

Comment 2: The first two points in the abstract should be simple for readers.

In the Abstract… Authors focused mostly on the background of macroalgae and need to write details about their findings also. Please rewrite the abstract.

Answer 2: More information was added in the abstract regarding the results obtained.

Comment 3: The introduction part needs to be improved towards connected the para to para.

Answer 3: The introduction was improved, according with the suggestions of the reviewers.

Comment 4: Can you please describe on what basis the authors selected the three examples? What is the current status of the three examples, in research?

Answer 4: It was added in the introduction the reason why the authors selected these 3 seaweed species. Their widespread distribution and their perennial character in the Atlantic coastal area were the major drivers they were chosen.

Comment 5: Did season really affected the content... however, you showed no significant differences in Figure 2. Did you check the quality? Please write.

Answer 5: The season had a minor impact in the polysaccharide content in red seaweeds and it is significant in the brown seaweed. Thus, the polysaccharides quality can be checked using FTIR-ATR analysis, as the bibliography supports. We added more information regarding the quality check in the discussion.

Comment 6: Please write clearly about your findings shown Figures 3, 5 & 7: In which season content analysis was done.

Answer 6: The FTIR-ATR spectra presented in the manuscript were performed in the autumn, and this detail was added into the materials and methods section 4.3.1. However, all of the spectra were identical between the seasonal samples.

Comment 7: In the discussion section, where you discussed the gut microbiome, please also write about the nutritional values of seaweeds in brief.

Answer 7: We added more information about the nutritional profile of the selected seaweeds.

Comment 8: The content benefits humans through various health benefits… as per the given title, what about the risk factors...??

Answer 8: It is highlighted in discussion in the penultimate paragraph before table 4 (as seaweed food intake) and after the table 5 (as isolated compound for isolated ingestion or in a processed product).

Comment 9: In this study, you restricted yourself up to FTIR analysis. Improve the discussion with other supportive research data.

Answer 9: We did not have the time not the necessary equipment to do further analyses. We added more substantive discussion about FTIR.

Comment 10: The outcome of the research in the abstract is not the same as that at the end of the Conclusion section. Please update. Conclude your findings and their future directions….

Answer 10: We re-wrote the conclusion and added more key information.

Reviewer 3 Report

Introduction: Sentence construction needs improvement. “various”, “due to”, ‘selected’, “during”, etc. were used extensively. Substantial rewriting is needed, as the narrated research story and argument were not convincing. The scientific urge for undertaking this research is missing.

The first sentence starts with “growing demands for various types of seaweeds” but instead of listing the driving demands, remaining part of the sentence includes details about how much of seaweed is being produced and how. Similarly, the second sentence on “concurrently, developing seaweed for wild stocks” has nothing to with the first sentence and overall research theme.

Third sentence of nutritional qualities, it might not be necessary to say only “selected seaweeds as sources of specialty products…”, and later only limiting to “diets in Asian countries”, and “, even until the present day” is unnecessary.

Likewise, rest of the sentences also need rewriting.

Polysaccharides cannot be referred as “compounds”

“antitumor, anticoagulant….antifungal” cannot qualify as food applications and will not be of interest to food industries.

“Thus, when seaweed polysaccharides are ingested…in order to achieve higher nutrient absorption rates”, logic and connectivity among the sentences is missing.

“Additionally, researchers have reported that agar, a polysaccharide present in some red seaweeds (agarophytes), can help in weight loss, through the ingestion of as little as a daily 4.5 g portion”, is another wrongly formatted statement. The previous sentence narrates the health benefits of algal polysaccharides, in that regard, there is no point to say “Additionally, researchers have reported that agar, a polysaccharide present in some red seaweeds”.

“…there are some concerns hindering some consumers to include them in their daily diet.” Not a well-constructed statement.

“Seaweeds not on these lists would be considered… human food supply chains”, not convincing and confusing.

All of a sudden in the next paragraph, seasonal variation on the polysaccharide content of seaweeds was mentioned as one of the authors research objectives. So far, there was no discussion on the effect of seasonal variation, in that sense, what is the basis to undertake this research?

“further assessed if the polysaccharide content meets the threshold established by the competent authorities…” in the previous paragraph it has already been mentioned FDA as the competent authority, thus rewrite the sentence.

In the end it was stated that “Moreover, the literature was reviewed in order to understand the positive and potentially negative effects of seaweed polysaccharides in human health”, why, what is the use. Such an analysis should be part of introduction.

Overall, Introduction needs substantial revision, with logical and convincing story development, along with the pressing reason(s) for undertaking the research.

Section 2.1.1: what was extracted, carrageenan or iota-carrageenan? Round the values to first decimal place, throughout the manuscript. Why there were seasonal variations on the carrageenan and extracts yield? In the discussion, this point was not discussed thoroughly.

FTIR spectra needs thorough analysis. All the listed bands need to associated with the corresponding vibration/bending/stretching.

“Agar differs from carrageenans as they have the L-configuration for the 4-linked galactose residue; nevertheless, they have some structural similarities with carrageenans”, this is already known to the reader, what is the use of this general statement?

Similarly, what is the use of “The main polysaccharide found in the brown alga (S. muticum) was alginic acid, a linear copolymer of mannuronic (M) and guluronic acid (G). Different types of alginic acid present different proportions and/or alternating patterns of different guluronic (G) and mannuronic (M) units”

The discussion section is not of much use, as aimlessly it goes on listing several known properties, which have nothing to do with the research reported in this submission. If interested, authors could publish them as a separate review article.

The submission title has nothing to do with the research reported in the rest of the draft.

Overall, sentence construction was poor, logical development was poor and research objectives were primitive that were selected without any scientific reason. For these reasons, this manuscript is recommended rejection and authors are encouraged to substantially revise the draft and submit for a fresh review process.

Author Response

Reviewer 3:

Comment 1: Introduction: Sentence construction needs improvement. “various”, “due to”, ‘selected’, “during”, etc. were used extensively. Substantial rewriting is needed, as the narrated research story and argument were not convincing. The scientific urge for undertaking this research is missing.

The first sentence starts with “growing demands for various types of seaweeds” but instead of listing the driving demands, remaining part of the sentence includes details about how much of seaweed is being produced and how. Similarly, the second sentence on “concurrently, developing seaweed for wild stocks” has nothing to with the first sentence and overall research theme.

Third sentence of nutritional qualities, it might not be necessary to say only “selected seaweeds as sources of specialty products…”, and later only limiting to “diets in Asian countries”, and “, even until the present day” is unnecessary.

Likewise, rest of the sentences also need rewriting.

Polysaccharides cannot be referred as “compounds”

“antitumor, anticoagulant… antifungal” cannot qualify as food applications and will not be of interest to food industries.

“Thus, when seaweed polysaccharides are ingested…in order to achieve higher nutrient absorption rates”, logic and connectivity among the sentences is missing.

“Additionally, researchers have reported that agar, a polysaccharide present in some red seaweeds (agarophytes), can help in weight loss, through the ingestion of as little as a daily 4.5 g portion”, is another wrongly formatted statement. The previous sentence narrates the health benefits of algal polysaccharides, in that regard, there is no point to say “Additionally, researchers have reported that agar, a polysaccharide present in some red seaweeds”.

“…there are some concerns hindering some consumers to include them in their daily diet.” Not a well-constructed statement.

“Seaweeds not on these lists would be considered… human food supply chains”, not convincing and confusing.

Suddenly in the next paragraph, seasonal variation on the polysaccharide content of seaweeds was mentioned as one of the authors research objectives. So far, there was no discussion on the effect of seasonal variation, in that sense, what is the basis to undertake this research?

“further assessed if the polysaccharide content meets the threshold established by the competent authorities…” in the previous paragraph it has already been mentioned FDA as the competent authority, thus rewrite the sentence.

In the end it was stated that “Moreover, the literature was reviewed in order to understand the positive and potentially negative effects of seaweed polysaccharides in human health”, why, what is the use. Such an analysis should be part of introduction.

Overall, Introduction needs substantial revision, with logical and convincing story development, along with the pressing reason(s) for undertaking the research.

Answer 1: The introduction was improved, according with the suggestions of the reviewer. Although, the “antitumor, anticoagulant….antifungal” are not keys for food industry, these properties are based in antioxidant activities from various seaweeds compounds, thus, these  properties are investigated by several European and North Americans  projects to enhance food quality and safety for human consumption.

“Moreover, the literature was reviewed in order to understand the positive and potentially negative effects of seaweed polysaccharides in human health”. Such an analysis should be part of introduction.” We consider it is most important to connect the parts in the discussion due to a specific analysis with the results obtained.

Comment 2: Section 2.1.1: what was extracted, carrageenan or iota-carrageenan? Round the values to first decimal place, throughout the manuscript. Why there were seasonal variations on the carrageenan and extracts yield? In the discussion, this point was not discussed thoroughly.

Answer 2: It was stated the extraction of carrageenan, and through FTIR-ATR analysis it was possible to assess that the type of carrageenan produced by this species was iota-carrageenan.

We added more discussion about this topic in the fourth paragraph of the discussion.

Comment 3: FTIR spectra needs thorough analysis. All the listed bands need to associated with the corresponding vibration/bending/stretching.

Answer 3: Three tables were added with the information requested, regarding each FTIR-ATR spectrum.

Comment 4: “Agar differs from carrageenans as they have the L-configuration for the 4-linked galactose residue; nevertheless, they have some structural similarities with carrageenans”, this is already known to the reader, what is the use of this general statement?

Answer 4: We highlighted this part to emphasize that whilst agar and carrageenan are similar polysaccharides, they are also quite different. Otherwise, those readers who have no knowledge of seaweeds, may think that carrageenan and agar are the same polymer.

Comment 5: Similarly, what is the use of “The main polysaccharide found in the brown alga (S. muticum) was alginic acid, a linear copolymer of mannuronic (M) and guluronic acid (G). Different types of alginic acid present different proportions and/or alternating patterns of different guluronic (G) and mannuronic (M) units”

Answer 5: Through the analysis and interpretation of the FTIR-ATR spectrum, the stated sentence highlights that other alginophytes could contain a different ratio of guluronic (G) and mannuronic (M) units; thus, the bioactivity could be affected.

Comment 6: The discussion section is not of much use, as aimlessly it goes on listing several known properties, which have nothing to do with the research reported in this submission. If interested, authors could publish them as a separate review article.

Answer 6: The aim of this study was to evaluate the seasonal polysaccharide yield of 3 widespread, relevant seaweed species for food and phycocolloid industries. Because of the paradigm regarding the health benefits/ risks of consumption of these polysaccharides (directly or indirectly), the authors decided to add information regarding both the potentially negative and the positive aspects.

Comment 7: The submission title has nothing to do with the research reported in the rest of the draft.

Answer 7: In the manuscript, we analyze a seaweed compound which at a certain level is considered a health promoter, however, the over-consumption or deficient extractions techniques (as demonstrated in the discussion) can provoke harmful effects in the human health. We added more information to this topic in the discussion.

Comment 8: Overall, sentence construction was poor, logical development was poor and research objectives were primitive that were selected without any scientific reason. For these reasons, this manuscript is recommended rejection and authors are encouraged to substantially revise the draft and submit for a fresh review process.

Answer 8: The authors improved the manuscript, according to the reviewer suggestions. The English was improved; the goal of the study was explained more explicitly.

The legislation is explicit regarding the application of these polymers as food additives; however, the legislation does not consider them when consumed in the whole seaweed. Due to increasing seaweed commercialization and consumption, and since polysaccharides represent almost half of the dried biomass, in our opinion, this parameter should be regulated.

Round 2

Reviewer 3 Report

It was a pleasure to read the revised draft. Certainly, authors have critically revised the draft and improved its quality and readability; it would have been more pleasing had if there were this meticulous at the beginning. Still there is room for improvement. A few suggestions are listed below that need to be addressed during the revision.

Lines 118-119: change to “…they are present round the year.”

Line 120: change “This study aimed to analyze the” to “Herein,”

Line 121: change to “…three different seaweed species of agarophyte, carrageenophyte and alginophyte.”

“Moreover” was extensively used throughout the manuscript. If possible, fix it to improve the readability.

Line 140: remove “carrageenan”

Table 1: “Sulfate esters” might be more appropriate. Instead of “Bound”, “chemical group” might be more suitable. Similarly, change “compound” to “letter code” . Follow same pattern in the rest of the tables.

Line 161: change “agar” to “its”

Make sure in all figure captions at the end “.” is added.

Line 200: “…were investigated…” in what? Be more specific and complete the sentence.

Lines 209-210: replace “this seaweed in order to obtain” with “for”

Line 212: change “authors” to “reports”

Line 214: change “fluctuated” to “fluctuate”

Line 216: remove “reported”

Line 217: add “but” before “with a maximum”

Line 243: change “their” to “the”

Lines 245-246: remove “present in at the wave number at 1220 cm-1”

Lines 249-250: change “those of presented by Pereira [61], and Pereira et al” to “literature reports”

Author Response

Reviewer 3:

Comment 1: It was a pleasure to read the revised draft. Certainly, authors have critically revised the draft and improved its quality and readability; it would have been more pleasing had if there were this meticulous at the beginning. Still there is room for improvement. A few suggestions are listed below that need to be addressed during the revision.

Lines 118-119: change to “…they are present round the year.”

Line 120: change “This study aimed to analyze the” to “Herein,”

Line 121: change to “…three different seaweed species of agarophyte, carrageenophyte and alginophyte.”

 “Moreover” was extensively used throughout the manuscript. If possible, fix it to improve the readability.

 Line 140: remove “carrageenan”

Table 1: “Sulfate esters” might be more appropriate. Instead of “Bound”, “chemical group” might be more suitable. Similarly, change “compound” to “letter code” . Follow same pattern in the rest of the tables.

Line 161: change “agar” to “its”

Make sure in all figure captions at the end “.” is added.

Line 200: “…were investigated…” in what? Be more specific and complete the sentence.

Lines 209-210: replace “this seaweed in order to obtain” with “for”

Line 212: change “authors” to “reports”

Line 214: change “fluctuated” to “fluctuate”

Line 216: remove “reported”

Line 217: add “but” before “with a maximum”

Line 243: change “their” to “the”

Lines 245-246: remove “present in at the wave number at 1220 cm-1”

Lines 249-250: change “those of presented by Pereira [61], and Pereira et al” to “literature reports”

Answer 1: We thank the kind word from the reviewer, and acknowledge the revision work done by him/her. Which improved greatly the manuscript quality. The authors improved the manuscript, according to the reviewer suggestions.